# α7 nicotinic acetylcholine receptor upregulation by anti-apoptotic Bcl-2 proteins

G. Brent Dawe[1], Hong Yu[1], Shenyan Gu[1], Alissa N. Blackler[1], Jose A. Matta[1], Edward R. Siuda[1,3], Elizabeth B. Rex[2,4] & David S. Bredt[1]

Nicotinic acetylcholine receptors (nAChRs) mediate and modulate synaptic transmission throughout the brain, and contribute to learning, memory, and behavior. Dysregulation of α7-type nAChRs in neuropsychiatric as well as immunological and oncological diseases makes them attractive targets for pharmaceutical development. Recently, we identified NACHO as an essential chaperone for α7 nAChRs. Leveraging the robust recombinant expression of α7 nAChRs with NACHO, we utilized genome-wide cDNA library screening and discovered that several anti-apoptotic Bcl-2 family proteins further upregulate receptor assembly and cell surface expression. These effects are mediated by an intracellular motif on α7 that resembles the BH3 binding domain of pro-apoptotic Bcl-2 proteins, and can be blocked by BH3 mimetic Bcl-2 inhibitors. Overexpression of Bcl-2 member Mcl-1 in neurons enhanced surface expression of endogenous α7 nAChRs, while a combination of chemotherapeutic Bcl2-inhibitors suppressed neuronal α7 receptor assembly. These results demonstrate that Bcl-2 proteins link α7 nAChR assembly to cell survival pathways.

[1] Neuroscience Discovery, Janssen Pharmaceutical Companies of Johnson & Johnson, 3210 Merryfield Row, San Diego, CA 92121, USA. [2] Discovery Sciences, Janssen Pharmaceutical Companies of Johnson & Johnson, 3210 Merryfield Row, San Diego, CA 92121, USA. [3] Present address: Alkermes, Inc., 852 Winter Street, Waltham, MA 02451, USA. [4] Present address: Janssen Scientific Affairs, LLC, 800 Ridgeview Drive, Horsham, PA 19044, USA. Correspondence and requests for materials should be addressed to D.S.B. (email: DBredt@its.jnj.com)

Nicotinic acetylcholine receptors (nAChRs) are pentameric ligand-gated ion channels that mediate synaptic transmission at the neuromuscular junction and cholinergic synapses in the brain and periphery[1]. The neuronal nAChR family comprises α2-10 and β2-4 subunits, which give rise to numerous receptor subtypes through selective heteromerization[2,3]. However, the α7 subunit is atypical in that it forms homomeric complexes[4] with five equivalent agonist-binding sites in the extracellular domain[5]. Native α7 receptors are expressed on both pre- and postsynaptic membranes, where they serve as modulators of circuit activity throughout the brain[3,6]. Though α7 nAChRs desensitize on a sub-millisecond time scale (e.g. ref. [7]), their relatively high calcium (Ca$^{2+}$) permeability[8] has physiological importance. Indeed, deletion or selective activation of α7 receptors shows that they facilitate learning and memory, and also contribute to anxiety[9,10]. Likewise, deficits in α7 nAChR function are implicated in Alzheimer's disease, schizophrenia, and pain, making the augmentation of its activity an attractive therapeutic strategy[11,12].

Recombinant expression of functional α7 nAChRs is poor in most mammalian cell lines[13–15], though we recently identified the transmembrane protein NACHO (TMEM35A) as critical for efficient expression of recombinant and native α7 nAChRs[16]. Furthermore, the activity of several other neuronal nAChR subtypes is also enhanced by NACHO[17]. Biochemical studies show that NACHO mediates α7 pentamer formation, as NACHO is required for receptor labeling by α-bungarotoxin (α-Bgt)[16], which binds at the interface between assembled α7 subunits[4,18]. Interestingly, resistance to inhibitors of cholinesterase-3 (Ric-3) protein, which is required for nAChR function in *Caenorhabditis elegans*[19], synergizes with NACHO to promote receptor assembly and function[16].

Here, we screened a genome-wide cDNA library to determine if additional proteins further augment α7 nAChR expression in the presence of NACHO. Unexpectedly, several members from the B-cell lymphoma 2 (Bcl-2) protein family were amongst the top hits. Bcl-2 proteins are widely known for regulating apoptosis, as the binding of antiapoptotic Bcl-2 proteins to proapoptotic Bcl-2 proteins prevents signaling cascades that initiate programmed cell death[20]. Because overexpression of antiapoptotic Bcl-2 proteins promotes unchecked cell proliferation, these molecules have proven effective targets for anticancer drugs[21]. Interestingly, the activation of α7 nAChRs is also associated with prosurvival signaling pathways (e.g. ref. [22]) and generally thought to protect against neuron loss in neurodegenerative disorders[23]. Though prosurvival Bcl-2 members are abundantly expressed in the nervous system[24], whether they directly interact with α7 subunits or those of other neuronal nAChRs to facilitate cell survival has remained poorly studied.

We now find that α7 nAChRs possess a motif on the intracellular loop between transmembrane (TM) segments TM3 and TM4 that is highly conserved with the Bcl-2 homology (BH) 3 domain of proapoptotic Bcl-2 proteins. Mutation of this BH3-like motif disrupts Bcl-2-mediated α7 nAChR upregulation, though the effects of NACHO and Ric-3 remain, suggesting Bcl-2 proteins exploit unique structural pathways to enhance assembly. Moreover, manipulation of Bcl-2 family member myeloid cell leukemia-1 (Mcl-1) in neurons regulates α7 nAChR assembly, implying a functional role for this pathway in the biogenesis of native α7 receptors.

## Results

### α7 nAChR surface expression is enhanced by Bcl-2 proteins.

We cotransfected HEK293T cells with individual plasmids encoding the α7 subunit, NACHO, and one other gene product from a genome-wide collection of over 17,000 cDNA clones[25]. High-content imaging of transfected cells was then used to identify cDNA clones encoding proteins that facilitate α7 nAChR expression. Based on augmentation of cell surface α-Bgt labeling, the first, second, and fourth most effective clones encoded antiapoptotic Bcl-2 proteins, namely Mcl-1, Bcl-2-like 2 (Bcl-W), and Bcl-2-like 1 (Bcl-X$_L$) (Table 1). For each of these clones, surface labeling of α7 receptors by α-Bgt was at least 70% of the maximal levels observed when NACHO and Ric-3 were coexpressed with α7 subunits (see ref. [17]), though no other Bcl-2 proteins present in the library appeared as hits. To further evaluate the effect of Bcl-2 members on α7 nAChRs, we performed additional staining and electrophysiology experiments on transiently transfected cells. As Mcl-1 and Bcl-X$_L$ are much more abundant in brain tissue, where functional α7 receptors occur, compared to Bcl-2 itself (Genotype Tissue Expression project; https://gtexportal.org), we focused on these two proteins for most of our studies.

When Mcl-1, Bcl-X$_L$, and Ric-3 cDNAs were cotransfected with α7 alone, total α-Bgt staining was barely, if at all detectable (Fig. 1a). However, coexpression of α7 subunits with NACHO yielded robust α-Bgt labeling in permeabilized and nonpermeabilized cells, as shown previously[16], which was further enhanced by each Bcl-2 protein (Fig. 1a). When α7, NACHO, and Bcl-2 member cDNAs were cotransfected at a 1:3:5 ratio, upregulation of surface staining was particularly evident, increasing 4.2 ± 0.3 fold (mean ± SEM; $n = 5$) or 2.9 ± 0.2 fold ($n = 5$) when Mcl-1 or Bcl-X$_L$ were respectively coexpressed

**Table 1 cDNAs that enhance surface α-Bgt staining of α7 nAChRs in HEK293T cells**

| Gene | Accession | Description | Cortex RNA | Effect vs. Ric-3 (%) |
|---|---|---|---|---|
| MCL1 | NM_021960.4 | Myeloid cell leukemia-1 | 43 | 84.4 |
| BCL2L2 | NM_004050.4 | Bcl2-like 2 | 102 | 83.1 |
| VEZT | NM_017599.3 | Vezatin | 10 | 75.7 |
| BCL2L1 | NM_138578.1 | Bcl2-like 1 | 54 | 73.3 |
| FAM78B | NM_001017961.3 | Family with sequence similarity 78 member B | 3.3 | 59.0 |
| *BCL2L1* | *NM_138578.1* | *Bcl2-like 1* | *54* | *55.2* |
| VDAC2 | NM_003375.3 | Voltage-dependent anion channel 2 | 66 | 54.9 |
| SLC25A43 | NM_145305.2 | Solute carrier family 25 member 43 | 3.2 | 51.5 |
| VDAC1 | NM_003374.2 | Voltage-dependent anion channel 1 | 129 | 51.4 |
| CHCHD6 | NM_032343.1 | Coiled-coil-helix-coiled-coil-helix domain containing 6 | 20 | 51.0 |

Cortical RNA expression values were obtained from the Genotype-Tissue Expression (GTEx) Project portal with values reported in transcripts per million, or TPM. The effect ($E$) of each cDNA on surface α-Bgt staining was calculated from mean fluorescence intensity values using the following equation: $E = $ (sample − baseline control)/(positive control − baseline control) × 100%, where baseline control refers to staining produced by α7 and NACHO cotransfection and positive control refers to staining produced by α7, NACHO, and Ric-3 cotransfection. Baseline and positive controls were specific to each plate that was assayed. A duplicate BCL2L1-containing cDNA construct present in the library is indicated by italicization

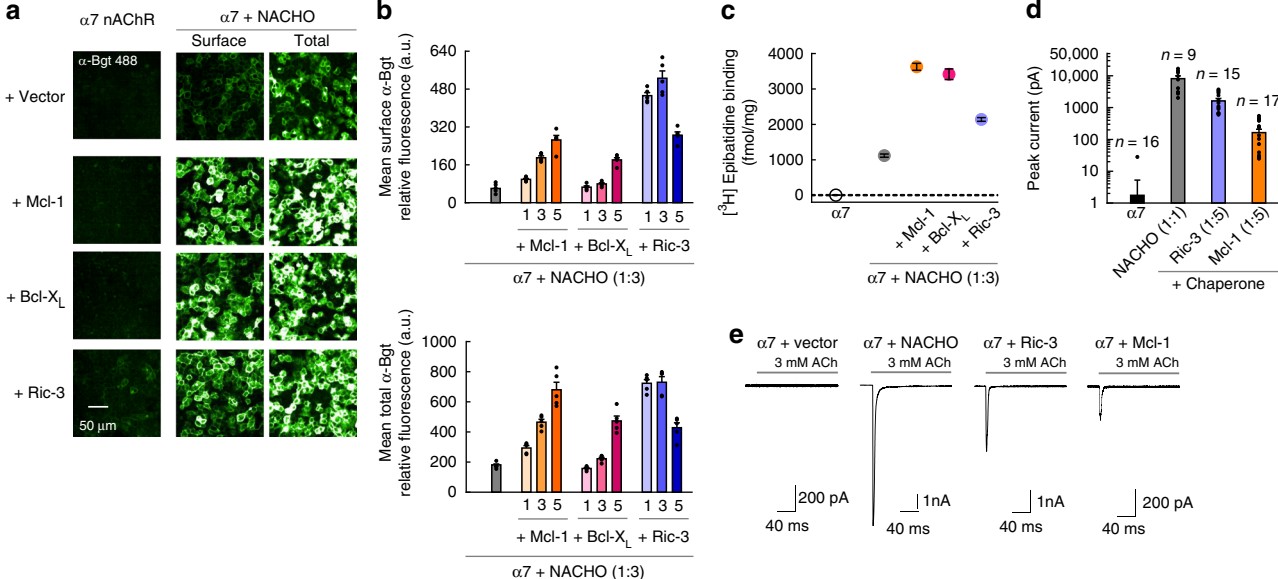

**Fig. 1** Bcl-2 proteins enhance assembly and functional expression of α7 nAChRs. **a** Immunofluorescent staining of HEK293T cells cotransfected with cDNAs encoding α7, or α7 and NACHO, along with Mcl-1, Bcl-X$_L$, or Ric-3 (at 1:3:5 cDNA ratio, respectively). Surface and total staining were performed before and after membrane permeabilization, respectively, using α-Bgt conjugated to a fluorescent probe. **b** Quantification of specific α-Bgt labeling, expressed as fluorescence intensity ($n = 5$). cDNA ratios are indicated relative to the α7-encoding vector. At 5× cDNA ratios, the increase in surface (top) and total (bottom) α-Bgt labeling produced by Mcl-1, Bcl-X$_L$, or Ric-3 cotransfection was significant ($p < 0.001$). The same cDNA combinations were transfected at different passage numbers and yielded similar results. **c** Binding of 10 nM [$^3$H]epibatidine to HEK293T cell membranes cotransfected with α7, NACHO, and other proteins as indicated (at 1:3:5 cDNA ratio, respectively), with measurements taken from multiple samples during a single experiment ($n = 6$). The increase in binding produced by Mcl-1, Bcl-X$_L$, or Ric-3 cotransfection atop α7 and NACHO was significant ($p < 1e^{-5}$). **d, e** Summary graph of agonist-evoked peak currents for HEK293T cells cotransfected with specified cDNAs along with eGFP, with $n$ values listed for each condition. Versus α7 expressed alone, where only 1 of 16 cells responded to ACh, coexpression of either NACHO, Ric-3, or Mcl-1 yielded significant effects ($p < 0.01$). Ratios of each cDNA are indicated relative to the α7-encoding vector. Representative whole-cell current responses elicited from the conditions in panel (**d**) are also shown (**e**). All data are means ± SEM; $p$ values from two-sample $t$ test

with NACHO (Fig. 1b; Supplementary Fig. 1a), and Bcl-W or Bcl-2 itself produced comparable effects (Supplementary Fig. 1b). By comparison, Ric-3 produced a 4.6 ± 0.2 fold ($n = 5$) increase in α-Bgt surface staining when coexpressed with NACHO (Fig. 1b), while Bcl-2 protein coexpression with NACHO and Ric-3 further enhanced staining (Supplementary Fig. 1c). Because Bcl-2 proteins elicited comparable increases in total versus surface α-Bgt staining (Supplementary Table 1), these data suggest intracellular assembly of α7 subunits was enhanced, rather than merely membrane trafficking. In agreement, membrane binding of the nAChR agonist [$^3$H] epibatidine, which requires assembly of two subunits to form the orthosteric site, also increased to similar extents when Mcl-1, Bcl-X$_L$, and Ric-3 were cotransfected with α7 and NACHO cDNA transcripts (Fig. 1c; Supplementary Fig. 1d). Moreover, Bcl-2 protein-dependent increases in total α-Bgt staining persisted in the presence of the proteasome inhibitor MG-132, and did not occur alongside comparable increases in α7 subunit expression (Supplementary Fig. 1e, f; Supplementary Table 1). This indicates that Bcl-2 member effects on α7 nAChR assembly do not stem from a suppression of ubiquitination and/ or degradation of unassembled subunits.

Given that detectable electrophysiological responses have been obtained from mammalian cells when Ric-3 is coexpressed with α7 subunits (e.g. ref. [16]), despite minimal α-Bgt staining, we wondered whether the expression of Bcl-2 proteins could be sufficient to produce α7-mediated currents. When α7 and Mcl-1 cDNAs were cotransfected at a 1:5 ratio, whole-cell ACh-evoked currents were routinely observed, with peak amplitudes averaging 163 ± 46 pA ($n = 17$), as opposed to just 1

of 16 cells transfected with α7 alone (Fig. 1d, e). Larger α7 peak currents were recorded in the presence of Ric-3 (1.6 ± 0.3 nA, $n = 15$), or especially NACHO (8.1 ± 1.8 nA, $n = 9$). Cotransfecting Bcl-2 member cDNAs with NACHO further increased α7 peak currents (Supplementary Fig. 2). Taken together, these findings reaffirm that Bcl-2 proteins and NACHO can work in synergy to enhance functional expression of α7 nAChRs.

**Bcl-2 inhibitors disrupt Bcl-2-mediated α7 upregulation.** Interactions between antiapoptotic Bcl-2 proteins and their proapoptotic counterparts occur through a binding groove formed by BH domains 1–3 of the former group[26]. Chemotherapeutic inhibitors, including the apoptosis-inducing molecule ABT-737[27], occupy this groove[28], which liberates the proapoptotic BH3 proteins (Supplementary Fig. 3a, b). More recently, the compounds S-63845 (S63 [29]) and A-1155463 (A11 [30]) have been reported to selectively inhibit Mcl-1 and Bcl-X$_L$, respectively, at nanomolar concentrations. Incubation for 24 h with 100 nM S63 reduced Mcl-1 upregulation of surface α-Bgt staining from 4.9 ± 0.2 fold ($n = 5$) to 2.3 ± 0.1 fold ($n = 5$) above that of α7 + NACHO alone (Fig. 2a, b). Likewise, incubation for 24 h with 100 nM A11 abolished Bcl-X$_L$, but not Mcl-1 or Ric-3-mediated enhancement of surface α-Bgt staining (Fig. 2a, b), resulting in staining intensity that decreased from 3.6 ± 0.2 fold ($n = 5$) to 0.92 ± 0.04 fold ($n = 5$) above that of α7 + NACHO alone. Electrophysiological data also showed that A11 strongly suppressed α7 nAChR currents obtained in the presence of Bcl-X$_L$ (Fig. 2c, d). In the micromolar range, ABT-737 countered the effect of Bcl-2 proteins on α7 nAChR surface expression (Supplementary Fig. 3c, d).

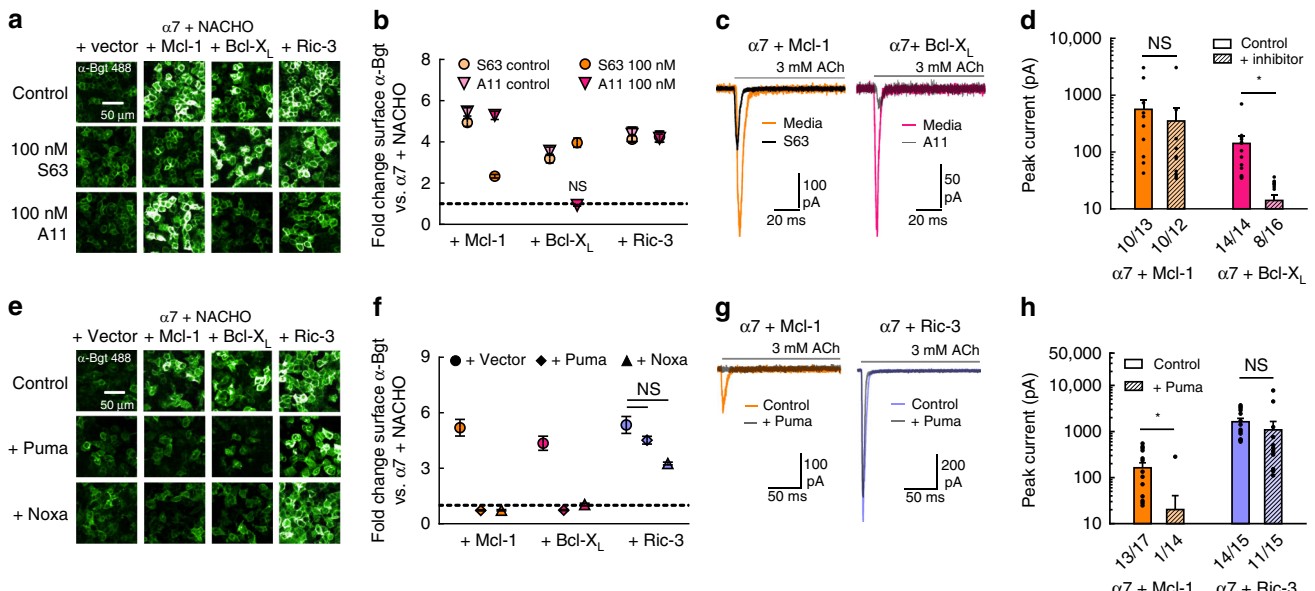

**Fig. 2** Chemical inhibitors and BH3-only proteins eliminate Bcl-2-mediated upregulation of α7 nAChRs. **a** Fluorescent α-Bgt labeling of nonpermeabilized HEK293T cells cotransfected with cDNAs encoding α7 and NACHO, along with other proteins indicated, at a 1:3:4 respective ratio. **b** Surface α-Bgt labeling with S63 or A11 inhibitors, presented as a fold change in fluorescence intensity ($n = 5$ treated, $n = 10$ control), relative to α7 and NACHO expressed alone. Enhancement of fluorescence remained significant in all cases ($p < 0.01$) except cells expressing Bcl-X$_L$ treated with 100 nM A11 ($p = 0.13$). **c, d** Representative whole-cell current responses elicited from HEK293T cells expressing α7, and either Mcl-1 or Bcl-X$_L$ (1:5 cDNA ratio), with or without 24 h incubation at 30 °C in the inhibitors A11 and S63 (1 μM), respectively (**c**). Summary graph of agonist-evoked peak currents (*$p < 0.01$) with the number of responsive cells indicated (**d**). The reduction in peak response was significant for A11, but not S63 ($p = 0.56$) treatment. **e** Fluorescent α-Bgt labeling of nonpermeabilized HEK293T cells cotransfected with cDNAs encoding α7 and NACHO, along with other proteins indicated, including Puma or Noxa at a 1:3:4:4 (α7: NACHO: other: BH3) cDNA ratio. **f** Fold change in fluorescence intensity for α-Bgt labeled cells in the presence of BH3 only proteins ($n = 5$), relative to α7 and NACHO expressed alone. Coexpression of Puma or Noxa significantly reduced fluorescence intensity in the presence of Mcl-1 or Bcl-X$_L$ ($p < 0.01$), but not Ric-3 ($p ≥ 0.22$). **g, h** Representative whole-cell current responses elicited from HEK293T cells cotransfected with cDNAs encoding α7 and either Mcl-1 or Ric-3, ±Puma at a 1:5:5 respective cDNA ratio (**g**). Summary graph of agonist-evoked peak currents (*$p < 0.01$) with the number of responsive cells indicated (**h**). Control values taken from Fig. 1d. The reduction in peak response was significant for Mcl-1, but not Ric-3 ($p = 0.41$) expressing cells. All data are means ± SEM; $p$ values from two-sample $t$ test. For all α-Bgt labeling experiments, the same cDNA transfection conditions were repeated at different passage numbers and yielded similar results

We next asked whether the overexpression of proapoptotic BH3-only proteins would also block Bcl-2 protein-mediated α7 nAChR upregulation. The cotransfection of either p53-mediated upregulator of apoptosis (Puma) or Noxa cDNA atop α7 and NACHO prevented Mcl-1 and Bcl-X$_L$ from enhancing surface α-Bgt staining (Fig. 2e). Specifically, the coexpression of Mcl-1 or Bcl-X$_L$ with Puma resulted in staining intensities that were 72 ± 3% ($n = 5$) and 74 ± 3% ($n = 5$) of that obtained without them, whereas Ric-3 increased staining 4.5 ± 0.2 fold ($n = 5$; Fig. 2f). Furthermore, Puma expression largely abolished ACh-evoked currents when Mcl-1, but not Ric-3 was cotransfected with α7 cDNA (Fig. 2g, h). Noxa also blocked Mcl-1 and Bcl-X$_L$ enhancement of α7 nAChR surface expression, as staining intensities were 73 ± 4% ($n = 5$) and 104 ± 8% ($n = 5$) of control values, though Ric-3 increased staining 3.3 ± 0.1 fold ($n = 5$; Fig. 2e, f). Overall, these data suggest the binding groove of antiapoptotic Bcl-2 members, typically engaged by proapoptotic BH3 motifs, is important for upregulation of α7 nAChR assembly.

**α7−Bcl-2 interactions depend on a BH3-like binding motif.** To assess the role for the Bcl-2 binding groove on α7 nAChR assembly, we explored whether that region is sufficient for Mcl-1-mediated upregulation, and how some segment of the α7 protein could act as a ligand there. The topology of antiapoptotic Bcl-2 proteins is largely cytosolic, and a C-terminal TM domain anchors them to mitochondrial and endoplasmic reticulum (ER)

membranes (Fig. 3a). Deletion of the TM region of Mcl-1 disrupted its effects on α7 subunit assembly. However, replacement of the TM segment with that of the ER resident protein cytochrome B5 (CytB5) restored the upregulation of α7 pentamer formation, producing comparable effects to wild-type Mcl-1 (Fig. 3b–d). Taken together, these data demonstrate that a membrane tether is required to bring Bcl-2 proteins into proximity with α7 subunits and that their interaction likely occurs in the ER (see Discussion).

To identify regions on the α7 subunit that might interact with Bcl-2 proteins we focused on the intracellular loop between TM3 and TM4, which comprises the vast majority of the cytoplasmic portion of the protein. Interestingly, a helical motif prior to TM4 contains an amino acid sequence with striking similarity to the BH3 domain of proapoptotic Bcl-2 proteins (Fig. 4a). Hydrophobic residues are in fact conserved in α7 at three key positions known to mediate BH3 affinity for the Bcl-2 binding groove (e.g. ref. [31]) (Fig. 4b). We therefore tested various deletions and mutations in this vicinity.

Deletion of the entire α7 TM3−TM4 intracellular loop prevents subunit assembly, yet removal of most of the loop still permits assembly-dependent α-Bgt binding[32]. One such mutant (del 347) yielded α-Bgt labeled receptors (with NACHO present) whose expression was still enhanced by Ric-3 (2.7 ± 0.2 fold, $n = 5$), whereas enhancement by Mcl-1 and Bcl-X$_L$ was markedly reduced to 1.0 ± 0.1 fold ($n = 5$) and 1.2 ± 0.03 fold ($n = 5$), respectively (Fig. 4c, d). More targeted removal of the helical

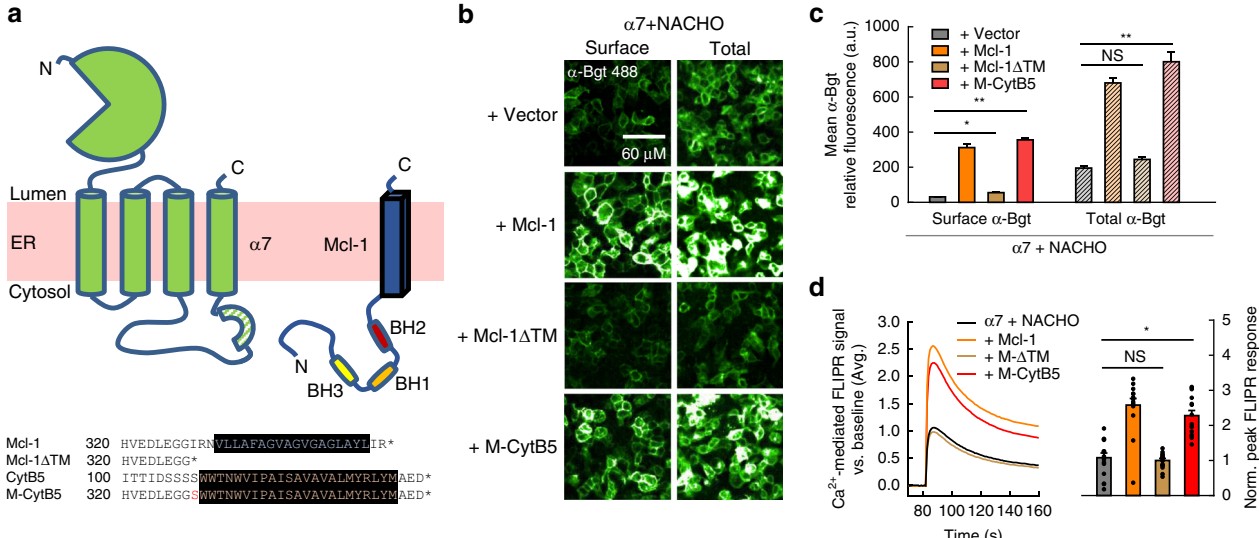

**Fig. 3** ER-anchoring of the Mcl-1 TM segment is critical for regulation of α7 nAChR assembly. **a** Topological illustration of the α7 nAChR subunit and Mcl-1 within the ER membrane (top), and amino acid sequence alignment of various Mcl-1 constructs examined with transmembrane regions highlighted in black (bottom). **b** Fluorescent α-Bgt labeling of nonpermeabilized and permeabilized HEK293T cells cotransfected with cDNAs encoding α7 and NACHO, along with empty vector, wild-type Mcl-1, a truncated Mcl-1 construct lacking the C-terminal TM segment (Mcl-1ΔTM), or a chimeric Mcl-1 containing the TM sequence of cytochrome B5 in place of its own (M-CytB5) at a 1:3:4 respective ratio. **c** Quantification of fluorescence intensity from surface and total labeling of α7 nAChRs by α-Bgt when coexpressed with NACHO and various Mcl-1 constructs ($n = 5$, *$p < 0.01$, **$p < 0.001$). Increases in total α-Bgt labeling were significant ($\alpha = 0.017$) for cells expressing Mcl-1 and M-CytB5, but not Mcl-1ΔTM ($p = 0.04$), atop α7 and NACHO. The same cDNA combinations were transfected at different passage numbers and yielded similar results. **d** Averaged FLIPR traces showing 100 μM nicotine-evoked $Ca^{2+}$ flux through HEK293T cells cotransfected with cDNAs encoding the α7 nAChR, NACHO, and various Mcl-1 constructs at a 1:1:5 respective ratio (left). Also shown is the mean peak response above baseline ($n = 15$, *$p < 1e^{-5}$), normalized to the background fluorescence signal (right). Increases in FLIPR response were significant for cells expressing Mcl-1 and M-CytB5, but not Mcl-1ΔTM ($p = 0.60$), atop α7 and NACHO. Data shown are from an individual experiment that was replicated. All data are means ± SEM; $p$ values from two-sample $t$ test

region surrounding the BH3-like motif (del preM4) produced similarly attenuated Bcl-2 protein upregulation of total α-Bgt staining, as did point mutations of hydrophobic amino acids at positions 433, 436, and 440 to alanine residues (Fig. 4c, d). The effect of these mutations was also consistent in experiments measuring α-Bgt staining of nonpermeabilized cells (Supplementary Fig. 4a, b). All these pre-M4 helix manipulations also greatly increased surface and total α-Bgt staining in the control condition, where only α7 subunits and NACHO were coexpressed. Such findings mirror earlier reports of mutations in this region augmenting surface α-Bgt binding in α7-expressing oocytes[33]. The possibility that greater basal receptor expression simply precludes further upregulation by accessory proteins is countered by the L433A mutation, which yielded the highest baseline α-Bgt labeling of all mutant receptors examined yet experienced a 1.9 ± 0.1 fold ($n = 5$) increase in total staining with Ric-3 cotransfected (Fig. 4d). A similar effect was seen in electrophysiological data, where α7 L433A yielded consistent current responses without any chaperone protein coexpressed (Supplementary Fig. 4c, d). As a result, the most likely explanation for the reduced effect of Bcl-2 proteins on mutant receptors is a selective disruption of their interaction with α7 subunits. Though pre-M4 helix point mutations reduced Ric-3 enhancement of α7 nAChR surface expression, the chaperone still consistently produced two-fold or greater increases in total α-Bgt labeling, more than any residual effect of either Mcl-1 or Bcl-X$_L$ (Fig. 4).

To verify that the limited effect of Bcl-2 proteins on mutant α7 nAChR surface expression reflects changes in functional receptor populations, we studied the electrophysiological behavior of several mutants. Given that agonist-evoked, $Ca^{2+}$-mediated fluorescence imaging plate reader (FLIPR) responses from the

del 347 and del preM4 mutants were absent or negligible, relative to those produced by wild-type α7 nAChRs (Supplementary Fig. 4e), we concluded that severe perturbations of the pre-M4 helix are detrimental for channel activity. Accordingly, we focused on the I436A point mutation and performed functional recordings in the absence of NACHO, though transfected cells were incubated at 30 °C to enhance receptor surface expression (e.g. ref. [34]). Consistent with α-Bgt staining, coexpression of Mcl-1 dramatically enhanced wild-type α7 peak currents over ten-fold from 34 ± 15 pA ($n = 22$) to 565 ± 256 pA ($n = 13$), whereas I436A peak currents were only modestly and non-significantly increased from 392 ± 97 pA ($n = 17$) to 709 ± 382 pA ($n = 9$; Fig. 4e, f).

**Regulation of α4β2 nAChRs by Bcl-2 proteins**. Another interesting facet of the α7 I436A mutation is that the equivalent position in most neuronal nAChR subunits contains an alanine residue (Supplementary Fig. 5a). As a result, one would expect any potential enhancement of receptor assembly by Bcl-2 proteins to be lessened by the reduced hydrophobicity of this position. To test this idea, we examined α4β2 receptors since they constitute the most abundant nicotinic subtype in the brain[3]. Staining of nonpermeabilized HEK293T cells cotransfected with α4 and a C-terminal, HA-tagged β2 construct revealed the presence of surface α4β2 receptors, because neither subunit expresses on the cell membrane when transfected alone (e.g. ref. [35]). Cotransfection of Bcl-X$_L$, along with α4, β2-HA, and NACHO modestly increased surface HA staining by 1.2 ± 0.2 fold ($n = 4$), whereas Mcl-1 yielded no apparent increase (0.9 ± 0.1 fold, $n = 4$; Supplementary Fig. 5b–d). Whole-cell peak current responses to ACh were also increased from 327 ± 67 pA ($n = 26$) to 693 ± 242 pA ($n = 18$) when α4β2 was coexpressed with

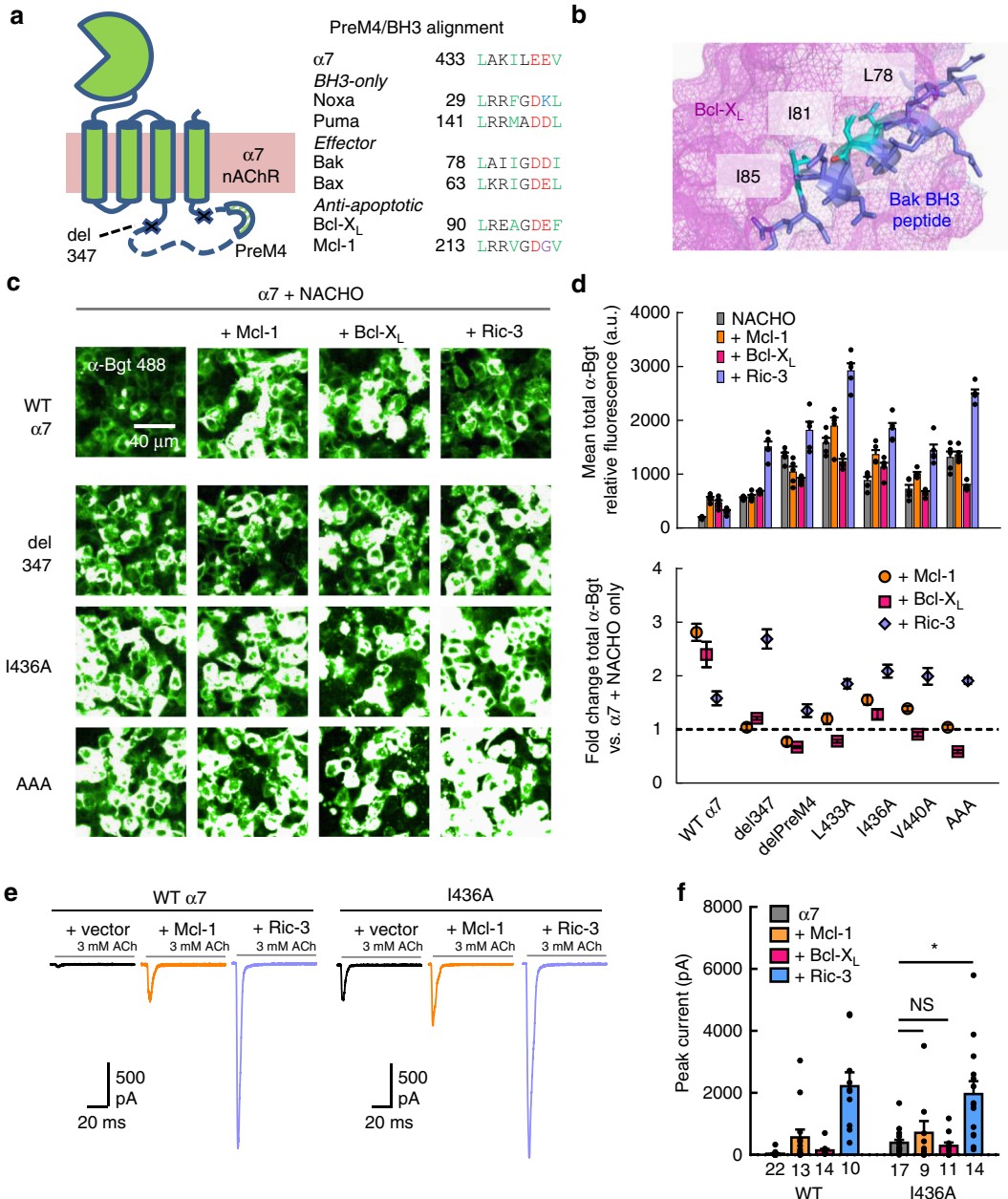

**Fig. 4** Mutations in a BH3-like motif of α7 nAChRs attenuate Bcl-2-mediated upregulation. **a** Cartoon illustration of α7 nAChR topology, including the region removed in the del 347 mutant and location of the pre-M4 helix (left). Also shown is a sequence alignment between the pre-M4 helix and BH3 domains of several Bcl-2 family proteins (right), color-coded by hydrophobicity, where hydrophobic (green), acidic (red), and basic (blue) residues at key positions are indicated. **b** Structure of Bcl-X_L bound by the BH3 segment of Bak (PDB: 1BXL[31]) with hydrophobic residues on Bak mediating the interaction highlighted (teal). **c** Fluorescent α-Bgt labeling of permeabilized HEK293T cells cotransfected with cDNAs encoding wild-type or mutant α7 and NACHO, along with other proteins, at a 1:3:4 respective ratio. **d** Quantification of fluorescence intensity from α7 mutants labeled by α-Bgt (top) and the fold change in fluorescence intensity relative to α7 and NACHO expressed alone (bottom; $n = 5$). The fold effect of Mcl-1 and Bcl-X_L coexpression was significantly reduced when comparing wild-type with mutant α7 subunits ($p < 0.01$), though Ric-3-mediated upregulation did not differ significantly for any mutant except del 347 (which was further enhanced). The same cDNA combinations were transfected at different passage numbers and yielded similar results. **e** Representative whole-cell current responses elicited from HEK293T cells transfected with cDNAs encoding wild-type or mutant α7 receptors, alone or with Mcl-1, or Ric-3 at a 1:5 (α7: other) cDNA ratio. Cells were incubated 24 h at 30 °C prior to recording. **f** Summary graph of agonist-evoked peak currents for transfections described in panel (**e**) (*$p < 0.01$) with the $n$ value for each condition indicated. Values for wild-type α7 with Mcl-1 and Bcl-X_L taken from data set in Fig. 2d. For the wild-type receptor, but not the I436A mutant ($p = 0.03$, WT; $p = 0.22$, I436A; two-sample one-tail $t$ test), Mcl-1 significantly increased peak currents. All data are means ± SEM; $p$ values from two-sample $t$ test

Bcl-X_L, though Ric-3 coexpression yielded greater current amplitudes ($1178 \pm 239$ pA, $n = 22$) and Mcl-1, in contrast, actually reduced functional expression ($115 \pm 31$ pA, $n = 18$, Supplementary Fig. 5e, f).

If the modest effects of Bcl-X_L on α4β2 nAChRs were due to interactions with the pre-M4 helix, we reasoned that they could be enhanced by mutating the alanine residues at positions 567 (α4) and 427 (β2) to isoleucine, as found in the α7 subunit.

Likewise, deletion of the pre-M4 helix should prevent any upregulation. Expressed on their own, α4 A567I/β2 A427I mutant receptors exhibited reduced surface staining compared to wild-type receptors, though the effect of Bcl-2 protein coexpression was increased in comparison to wild-type α4β2 (Supplementary Fig. 5b–d). The pre-M4 deletion meanwhile yielded much greater basal α4β2 surface expression, while remaining insensitive to Bcl-2 proteins, a similar effect to that observed for the equivalent deletion in α7 subunits (Supplementary Fig. 5b–d).

**Mcl-1 can regulate neuronal α7 nAChR assembly.** We next explored whether endogenously expressed α7 nAChRs can be regulated by Bcl-2 proteins, similar to recombinant receptors. Accordingly, we transduced cultured hippocampal neurons with lentivirus encoding Mcl-1, and assayed α7 nAChR surface expression using fluorescent α-Bgt staining and $Ca^{2+}$-mediated FLIPR responses. Mcl-1-specific staining intensity essentially doubled among neurons at multiplicity of infection (MOI) 100, versus nontransduced controls (Supplementary Fig. 6a), indicating that widespread recombinant expression had occurred. Given the relatively low expression of α7 nAChRs in cultured neurons, we performed experiments around DIV 20 when protein levels are maximal (see ref. [36]). Surface α-Bgt staining increased roughly twofold in neurons transduced at MOI 30 or 100 (Fig. 5a, b). As a positive control, incubation of neurons in 100 μM nicotine over several days produced robust upregulation of the α-Bgt signal, as described previously for cultured neurons (Fig. 5a, b) (e.g. ref. [37]). In contrast, neither surface staining of the AMPA-type glutamate receptor subunit GluA1 nor the mean number of nuclei significantly changed in Mcl-1-transduced neurons (Fig. 5a, b;

Supplementary Fig. 6b). This demonstrates that increases in α-Bgt staining are not likely to reflect a general enhancement of protein biogenesis or cell health in the presence of an antiapoptotic signaling factor.

Mcl-1 transduction also brought about a twofold increase in the nicotine-evoked FLIPR signal recorded from cultured neurons, comparable to the effect of preincubation in the same agonist (Supplementary Fig. 6c, d). These FLIPR responses were dependent on the presence of the α7-specific-positive allosteric modulator (PAM) PNU-12059[6][15] and blocked by inclusion of the competitive antagonist methyllycaconitine (MLA) (Supplementary Fig. 6c), demonstrating that they were in fact mediated by α7 nAChR activity.

Several antiapoptotic Bcl-2 members are abundantly expressed in the brain[38,39] and they may regulate the surface expression of neuronal α7 nAChRs. We addressed this possibility with the A11 and S63 compounds, which inhibit BH3 domain binding to Bcl-$X_L$ and Mcl-1, respectively. Because these compounds are toxic at high concentrations, we co-applied each inhibitor at 10 nM, a concentration that mitigated neuronal toxicity (Supplementary Fig. 6b), but still reduced Bcl-2-mediated enhancement of α7 nAChR assembly in transfected HEK293T cells (Supplementary Table 1). Used at these concentrations, together with nicotine to augment α7 receptor assembly, the BH3 mimetic compounds significantly reduced upregulation of surface α-Bgt staining intensity from 262 ± 54% to 156 ± 43% ($n = 4$) of control levels. Meanwhile surface GluA1 staining of nicotine-treated neurons was not significantly changed by Bcl-2 inhibitors ($n = 4$) (Fig. 5c, d).

In addition to neurons, functional α7 nAChRs have also been reported in astrocytes (e.g. refs. [40,41]). We therefore tested

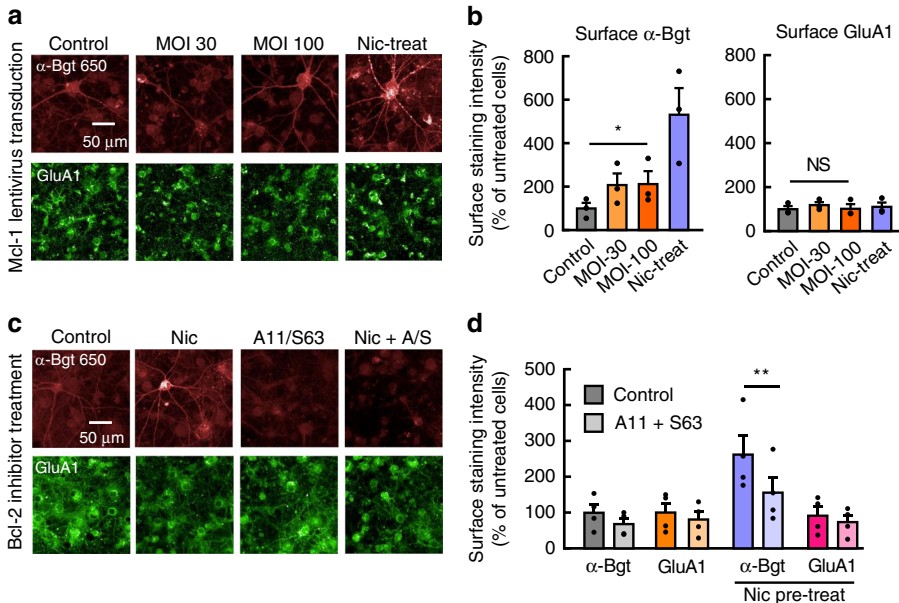

**Fig. 5** Bcl-2 proteins regulate α7 nAChRs in hippocampal neurons. **a** Fluorescent labeling of nonpermeabilized rat hippocampal neurons (DIV 20) transduced with lentivirus encoding human Mcl-1 at DIV7 or incubated with 100 μM nicotine (Nic) since DIV 15. Representative images of surface α-Bgt and surface GluA1 (AMPA receptor) labeling of neurons are shown. **b** Quantification of surface α-Bgt (left) and surface GluA1 (right) labeling, expressed as a percentage of the intensity from untreated neurons. Two multiplicities of infection (MOI 30 and 100) were compared to ensure maximal lentiviral transduction occurred. Data are averaged from three experiments on different cultures (*$p < 0.05$). At MOI 100, there was a significant increase in α-Bgt staining, but not GluA1 staining ($p = 0.41$, paired one-tail t test). **c** Fluorescent labeling of nonpermeabilized rat hippocampal neurons (DIV 21) treated for 96 h with the Bcl-$X_L$ inhibitor A-1155463 (A11, 10 nM) and Mcl-1 inhibitor S-63845 (S63, 10 nM), with or without coincubation in 100 μM nicotine (Nic) to promote α7 nAChR surface expression. Representative images of surface α-Bgt and surface GluA1 labeling are shown. **d** Quantification of surface α-Bgt and surface GluA1 fluorescence labeling intensity, expressed as a percentage of the fluorescence from untreated neurons. Data are averaged from four experiments on different cultures (**$p < 0.01$). With nicotine pretreatment, Bcl-2 inhibitors induced a significant decrease in α-Bgt staining, but not GluA1 staining ($p = 0.09$, paired one-tail t test). All data are means ± SEM

whether these receptor populations may be similarly regulated by Bcl-2 proteins. Incubation with the Bcl-2 inhibitors A11, S63, and ABT-737 reduced the nicotine-evoked FLIPR signal from cultured astrocytes, versus untreated cells (Supplementary Fig. 6e, f), suggesting Bcl-2 proteins can regulate α7 nAChR expression in multiple cell types where they are endogenously coexpressed.

## Discussion

This study advances our understanding of nAChR biology in several fundamental ways. First, we discover antiapoptotic Bcl-2 proteins can upregulate the assembly and surface expression of α7 nAChRs. In the presence of the essential α7 chaperone NACHO, several Bcl-2 members consistently enhanced α-Bgt labeling of assembled receptors to an extent comparable with Ric-3. Yet even without NACHO, coexpression of Bcl-2 proteins enabled reliable current responses to be detected from α7 subunits that would otherwise exhibit no functional activity. Second, we demonstrate that α7 nAChRs possess a BH3-like motif necessary for their regulation by antiapoptotic Bcl-2 proteins. This motif is situated within a helical segment of the α7 TM3-TM4 intracellular loop, and mutating this site eliminates Bcl-2 member-mediated, but not Ric-3-mediated receptor upregulation. Finally, we provide evidence that neuronal α7 nAChR activity is influenced by the presence of Bcl-2 proteins. In this regard, overexpression of Mcl-1 in cultured neurons produced a twofold increase in surface α-Bgt labeling and $Ca^{2+}$-mediated FLIPR responses. Overall, our data provide the first evidence for ion channel biogenesis to be regulated through molecular interactions with Bcl-2 proteins.

Although the α7 M3-M4 intracellular loop comprises nearly one-third of the mature protein, deleting most of this domain does not abolish channel function[32]. This suggests the intracellular region might serve as a substrate for other regulatory proteins. Interestingly, the amphipathic pre-M4 helix has been speculated as an interaction site for Ric-3, as mutations in that segment were found to reduce Ric-3-mediated upregulation of α-Bgt binding[42]. Our data, however, indicate Ric-3 can still facilitate assembly with the same residues absent in larger deletions, meaning that its interaction with α7 subunits does not depend solely on the pre-M4 region. Another interesting facet of the pre-M4 helix is that point mutations along the motif can enhance α7 nAChR expression in oocytes[33]. We confirmed this effect in mammalian cells. Particularly interesting is pre-M4 residue 436, which is alanine in many nAChR subunits like α2-4, β2, and β4, capable of assembly in HEK293T cells, but is isoleucine in α7. Remarkably, the α7 I436A mutant assembles much more readily in HEK293T cells, while losing most upregulation by Bcl-2 proteins. Because alanine at that position can be tolerated, it might seem counterintuitive for α7 subunits to retain a residue that hinders their assembly, though regulation by Bcl-2 protein chaperones may explain this idiosyncrasy. A partial snapshot of the pre-M4 helix is resolved in the *Torpedo* muscle nAChR[43], where the helices of each subunit become increasingly close as distance from the membrane increases. Because residues from one pre-M4 helix abut those of the adjacent subunit, it is conceivable that steric clashes within this site hinder assembly under normal circumstances. It has also been postulated that α7 subunits possess an ER retention motif (RFR) within residues 446–448[42], as the RXR motif is known to limit forward trafficking in other membrane proteins[44]. However, the pre-M4 deletion encompassing this area produced similar enhancements of α4β2 and α7 nAChR surface expression, despite the motif not being conserved outside the α7 subunit. This suggests that segments adjacent to the RFR sequence have a much greater influence on receptor trafficking. Accordingly, a plausible explanation for Bcl-2 family-mediated upregulation is not via

masking a retention signal, but rather through binding pre-M4 helices, holding apart the intracellular domains of the five subunits to facilitate assembly in an analogous manner as the pre-M4 deletion mutant.

The Bcl-2 protein family contains both antiapoptotic and proapoptotic/BH3 only members, which interact with each other through conserved BH1-4 domains[20]. Generally, the antiapoptotic proteins possess a hydrophobic groove formed by their BH1-3 segments that accommodates the helical BH3 ligand of proapoptotic binding partners[26]. Outside of the core Bcl-2 family, a growing number of unrelated, BH3 motif-containing proteins have been identified. While these noncanonical BH3-containing proteins are regulated by Bcl-2 proteins, they are not directly involved in apoptosis induction[45]. For example, the enzyme tissue transglutaminase[46], autophagy mediator Beclin-1[47], and transcription regulator SUFU[48] all possess BH3-like sequences through which Bcl-2 interactions regulate their activity. Bcl-2 and related proteins also interact with ryanodine and inositol tri-sphosphate ($IP_3$) receptor channels and inhibit their calcium release[49]. This regulation is independent of the BH domain binding groove and does not involve receptor assembly[50]. Indeed, Bcl-2 family proteins have not previously been found to mediate ion channel assembly. Ryanodine and $IP_3$ receptors reside in the ER, and Bcl-2 proteins are present in ER as well as mitochondria[51]. Interestingly, nAChRs undergo protein folding in the ER[52], and we find anchoring in the ER membrane is critical for Mcl-1-mediated upregulation of α7 subunit assembly (Fig. 3). Our experiments using Bcl-2 inhibitors to suppress α7 receptor expression on both neurons and astrocytes are also consistent with the idea that some Bcl-2 proteins localize to the ER in native systems.

Antiapoptotic Bcl-2 proteins control neuronal viability in a variety of different contexts[53–55]. Meanwhile, cholinergic signaling, particularly through α7 nAChRs, also promotes the survival and synaptic integration of adult-born neurons[22], leading to synergistic effects that maintain certain cell populations. For instance, the survival of tissue around ischemic brain lesions is notably enhanced by neuregulin-1[56,57], a highly variable (by alternative splicing) signaling protein, which is upregulated in injured cerebral tissue[58]. The treatment of neurons and microglia with exogenous neuregulin-1 increases α7 nAChR protein expression[59,60], though the mechanism of this effect is unclear. Because neuregulin-1 also increases transcript levels of Bcl-2 following ischemia[61], its effect on cell survival and α7 nAChR upregulation might stem from molecular interactions between Bcl-2 proteins and nicotinic receptors. Despite being beneficial for injured tissue, the antiapoptotic functions of Bcl-2 proteins can instead be problematic in cancer, where cell proliferation is unchecked by apoptosis. Nicotine and other nAChR agonists found in cigarette smoke are known to enhance Bcl-2 and Mcl-1 phosphorylation in lung cancer cell lines[62–64], as well as reduce ubiquitin-dependent Bcl-2 degradation[65], resulting in enhanced chemotherapeutic resistance. Furthermore, nicotine promotes tumor growth and metastasis in mouse models of lung cancer[66]. It is noteworthy that α7 nAChR activation can mediate Bcl-2 phosphorylation in certain instances above (i.e. ref. [63]), because nicotine also increases α7 subunit expression in lung cancer cells[67], which could provide a positive feedback mechanism. Enhanced Bcl-2 member expression could facilitate α7 subunit assembly and trafficking, leading to even more nicotine-mediated augmentation of Bcl-2 activity and the preservation of cells in an antiapoptotic state. Ultimately, more work will be needed to demonstrate a direct interaction between Bcl-2 proteins and α7 subunits during the physiological regulation of apoptosis, though our experiments demonstrate α7 nAChR expression is enhanced several fold when antiapoptotic Bcl-2 members are abundant.

## Methods

**Genes, molecular biology, and cell culture.** The genes studied here are all human forms: CHRNA7, variant 1 (NM_000746.5), CHRNA4, variant 1 (NM_000744.6), CHRNB2 (NM_000748.2); TMEM35A, or NACHO (NM_021637.2); Ric-3, variant 1 (NM_024557.5); BCL2, variant alpha (NM_000633.2); BCL2L1, or Bcl-X$_L$ (NM_138578.2); BCL2L2, or Bcl-W (NM_004050.4); MCL1, variant 1 (NM_021960.4); BBC3, or Puma, variant 4 (NM_014417.4); PMAIP1, or Noxa (NM_021127.2); CYB5A, variant 1 (NM_148923.3). Mutant proteins containing substitutions, insertions, and deletions were generated by site-directed mutagenesis with two complementary primer reactions run in parallel, and then confirmed by sequencing. Deletion constructs used in this study, with deleted residues in parentheses are: α7 del 347 (348–454), α7 del preM4 (432–450), α4 del preM4 (563–581), β2 del preM4 (423–441), Mcl-1 or M-ΔTM (328−). The M-CytB5 fusion construct was generated using primers with ends complementary to the cytosolic-transmembrane interface of Mcl-1, and a large central insert containing the CytB5 TM segment followed by a stop codon. The forward primer sequence was as follows: 5′-CAT GTA GAG GAC CTA GAA GGT GGC AGC TGG TGG ACC AAC TGG GTG ATT CCG GCG ATT AGC GCG GTG GCG GTG GCG CTG ATG TAT CGC CTG TAT ATG GCG GAA GAT TAG ATC AGG AAT GTG CTG CTG GCT TTT GC-3′. Other primers used to generate mutations can be found in a supplementary data file. The α7-HA and β2-HA constructs contained a PSGA linker and HA tag immediately following wild-type C-terminal residues A502 and K502, respectively. Puma and Noxa expression vectors were ordered from Origene, and encoded proteins were tagged by a C-terminal Myc-DDK tag. Residue numbering of all receptors includes the signal peptide. The Mcl-1-encoding Lentiviral vector contained the EF1α promoter for transgene expression, and viral particles were packaged by VectorBuilder. MOI was calculated based on a reported titer of $2.17 \times 10^6$ U/mL.

HEK293T cells used for recombinant expression assays were cultured in DMEM medium supplemented with 10% FBS and 1 mM sodium pyruvate. Transient transfections were performed using FuGENE® HD or FuGENE® 6 transfection reagents (Promega Corporation). Unless otherwise stated, assays were performed after incubating cells for 48–72 h at 37 °C following transfection.

Primary hippocampal neurons were prepared from E18 rat hippocampi (BrainBits). Dissociation was performed using digestion in 10 U/mL papain (Worthington) for 20 min, followed by trituration with a 10 mL glass pipette. The cell suspension was plated at 10,000 cells per well (384-well plates) and maintained in NbActiv4® media (BrainBits). Cultures were transduced with Mcl-1 encoding Lentivirus at 6–8 days in vitro (DIV) and surface staining or FLIPR assays were performed at 18–21 DIV, following 4–6 days incubation in nicotine for pretreated cultures. Astrocytes used for FLIPR assays were plated at 15,000 cells per well (384-well plates), after being isolated from mouse brains and cultured according to published methods[68].

**Broad cDNA library screen.** The cDNA library used for screening was purchased from the Broad Institute and contains 17,255 cDNA clones. Each cDNA was previously subcloned into the pLX-317 vector, and encodes a V5 tag at the C-terminus of the translated protein. The library was screened using 384-well plates, where each well was transfected with 60 ng of DNA, comprised of multiple expression vectors encoding α7 nAChR, NACHO, and a single Broad gene product at a 1:3:4 ratio. Surface α-Bgt labeling was assessed 48 h post transfection, when cells were incubated for 45 min at 37 °C with a fluorescent α-Bgt conjugate (AlexaFluor 647, Invitrogen™), prior to fixation in 4% paraformaldehyde (PFA) and imaging. A cDNA encoding a candidate α7 regulatory protein was classified as a hit if the increase in surface α-Bgt fluorescence intensity (averaged per cell) was greater than 31.0% of that induced by Ric-3 coexpression, used as a positive control. This threshold corresponded to three standard deviations from the mean change in α-Bgt intensity (−1.65%), resulting in a hit rate of 0.6%.

**Electrophysiology.** HEK293T cells were initially plated on uncoated plastic six-well plates, and 24 h prior to recording were treated for 5 min with CellStripper™ dissociation reagent (Corning) before being re-plated on 12 mm glass coverslips. During recording, cells were submerged in external solution composed of HyClone™ HEPES-buffered saline (GE Life Sciences) comprising 149 mM NaCl, 4 mM KCl, 10 mM HEPES, and 5 mM Glucose at pH 7.4 and 300 mOsm osmolality, supplemented with 2 mM CaCl$_2$ and 1 mM MgCl$_2$. Internal recording solution contained 145 mM CsCl, 2.5 mM NaCl, 10 mM HEPES, 4 mM MgATP, and 1.0 mM EGTA, with pH adjusted to 7.4 using CsOH. Osmolality was adjusted to 300 mOsm/kg using sucrose and measured with the Vapro osmometer (Wescor Inc.). Whole-cell recordings were made using the lifted-cell patch-clamp technique, with constant negative pressure applied into the recording pipette chamber to facilitate cell lifting (see ref. [69]). Recording pipettes were composed of borosilicate glass (World Precision Instruments) shaped to achieve 3–5 MΩ resistance using the PC-10 micropipette puller (Narishige). Agonist-evoked peak responses were measured by rapidly applying 3 mM acetylcholine (ACh) to eGFP-expressing cells, using the MXPZT piezoelectric actuator system (Siskiyou), at a membrane holding potential of −60 mV, unless otherwise indicated. Series resistances (3–15 MΩ) were routinely compensated by at least 80%. All recordings were performed using an Axopatch 200B amplifier (Molecular Devices). Signals were sampled at 25 kHz and low-pass filtered at 10 kHz. Data were acquired and analyzed using

pClamp10 software (Molecular Devices). Average peak current responses include data from GFP-expressing, nonresponsive cells, unless noted.

**Immunocytochemistry and image analysis.** Cells were either plated on 384-well BioCoat (Corning) or Cell Carrier Ultra clear-bottom microplates (PerkinElmer) that were precoated with poly-D-lysine. For HEK293T staining, fluorescent α-Bgt conjugates (2 μg/mL, AlexaFluor 488, Invitrogen™) or primary antibodies were incubated in culture media for 45 min at 37 °C prior to fixation and permeabilization to label surface proteins. All subsequent steps cells were performed in PBS at room temperature. In order, cells were fixed with 4% PFA for 60 min, permeabilized using 0.3% Triton X-100 (Sigma) for 15 min, and blocked in 10% normal goat serum for 30 min. For neuron staining, all steps were carried out with cells kept in HyClone™ HEPES-buffered saline (GE Life Sciences), described above, at room temperature. First, cells were fixed with 4% PFA for 60 min. Subsequently, fluorescent α-Bgt conjugates (1 μg/mL) or primary antibodies were incubated with cells for 60 min to enable surface staining, before simultaneous permeabilization and blocking in 0.2% Triton X-100 and 10% normal goat serum for 60 min. To label intracellular components, primary and secondary antibody incubation occurred sequentially for 1–2 h each. Prior to imaging, nuclei were stained with DAPI or NucBlue® reagent (Invitrogen™).

For quantitative imaging of stained cells, images were acquired using the Opera Phenix™ or Operetta™ screening system (PerkinElmer) with a 20× water objective. In every experiment five fields were acquired per well with Harmony™ high-content imaging software (PerkinElmer). Further analysis was done with the Columbus™ data storage and analysis system (PerkinElmer). Nuclei from viable cells were identified from the DAPI channel as having an area >30 μm$^2$ for HEK293T cells or an area >10 μm$^2$ for neurons. Cell boundaries were defined based on nuclear staining, and mean intensities were calculated from all regions of interest. Area of the cytoplasm was calculated based on the cytoplasm region of interest. Reported fluorescence intensities are averages of individually transfected or treated wells, where the mean intensity (across all identified cells) has been calculated, after subtracting the background signal from α-Bgt, HA, or antibody labeling of untransfected (HEK293T) or epibatidine cotreated (neurons) cells.

Primary antibodies used include Bcl-2 (Rb, 1:300, CST D17C4), Bcl-X$_L$ (Rb, 1:300, CST 54H6), Bcl-W (Rb, 1:300, CST 31H4), Mcl-1 (Rb, 1:300, CST D2W9E), HA (M, 1:500, Invitrogen™ 2-2.2.14) conjugated to DyLight 488 or DyLight 650, GluA1 N-terminal (M, 1:250, MilliporeSigma RH95), where M = mouse, Rb = Rabbit, and CST indicates Cell Signaling Technology. Secondary antibodies used include goat anti-mouse AlexaFluor 555 and goat anti-rabbit DyLight 650 (Invitrogen™) at 1:500 or 1:1000 dilution. In some cases, higher primary antibody concentrations were used, but kept constant within each experiment.

**Epibatidine binding assays.** Transfected HEK293T cells were harvested and resuspended in 50 mM cold TrisHCl buffer (pH 7.4). Cell membrane preparations were made by centrifugation at $33,000 \times g$ for 20 min at 4 °C. Membrane pellets were homogenized for 30 s using the T-25 Ultra-Turrax homogenizer (Ika). Protein concentrations were determined using the Pierce™ BCA Protein Assay Kit (Thermo Scientific) and normalized across samples prior to the assay. Binding assays were performed in 96-well plates, where samples were incubated with 10 nM (for α7) [³H]epibatidine for 3 h at room temperature. Nonspecific binding was determined by coincubation with 10 μM unlabeled epibatidine. Binding assays were terminated by filtration through polyethylenimine-treated Unifilter GF/B 96-well plates (PerkinElmer), which were immediately washed with 500 mL assay buffer per plate. Filter plates were desiccated at 65 °C for 20 min and incubated with 50 μL MicroScint-0 scintillant cocktail (PerkinElmer). Bound [³H]epibatidine was quantified using the TopCount NXT scintillation counter (PerkinElmer).

**FLIPR assay for Ca²⁺ influx.** High-throughput FLIPR assays were performed using 384-well BioCoat (Corning) plates. Assay buffer consisted of the HyClone™ HEPES-buffered saline (GE Life Sciences) described above, though prior to recordings, cells were loaded with Calcium 5 dye (Molecular Devices) for 1 h at room temperature. Plates were washed to remove excess dye, and then placed in the FLIPR Tetra (Molecular Devices) chamber. Recordings were captured using ScreenWorks 4.0™ software (Molecular Devices). Obtaining robust α7-mediated FLIPR responses required the presence of a selective PAM to attenuate desensitization, either 50 μM NS-1738 [70] for HEK293T cells or 5 μM PNU-120596 [15] for neurons. Neuronal FLIPR responses were also obtained in the presence of tetrodotoxin, or TTX (500 nM), to inhibit the firing of action potentials.

**Statistics.** Results are expressed as mean ± SEM. Analyses of sample means were performed using two-sample (assuming unequal variance) $t$ tests and significance was determined based on two-tailed $p$ values, unless otherwise indicated. The significance level $\alpha = 0.05$ was used, though for multiple comparisons the Bonferroni correction was applied, such that the significance threshold became $\alpha/n$, where $n$ is the number of comparisons.

**Reporting summary.** Further information on research design is available in the Nature Research Reporting Summary linked to this article.

## Data availability

The data that support the findings of this study are available from the authors upon request. Please see author contributions for specific data sets.

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

## Acknowledgements
The authors thank Yingbo He for the preparation of astrocytes.

## Author contributions
G.B.D., J.A.M. and E.R.S. performed electrophysiology experiments. G.B.D., H.Y. and S.G. performed immunocytochemistry experiments. A.N.B. and E.B.R. conducted cDNA library screening. G.B.D. and D.S.B. wrote the paper. All authors contributed to discussion and editing of the paper. D.S.B. supervised the project.

## Additional information

**Competing interests:** The authors declare no competing interests except that authors are all full-time employees of Janssen Pharmaceutical Companies of Johnson and Johnson.

