## [Peer Review File · Nature Communications]

Reviewers' comments:

Reviewer #1 (Remarks to the Author):

In manuscript "α7 nicotinic acetylcholine receptor upregulation by anti-apoptotic Bcl-2 proteins", the author reports findings suggesting that Bcl2 family proteins, mainly including Mcl-1 and Bcl-XL, regulate α7 nAChRs by their direct interaction via the BH3 domain in ER. These findings are interesting, which identified a potential new signaling pathway between Bcl2 family proteins and α7 nAChRs. However, the mechanisms studies in manuscript are weak and superficial, and lack real direct mechanisms. Authors always mentioned Bcl2 in the manuscript, it is hard to understand why there are no any experiments to actually test the effect of Bcl2 on α7 expression in this manuscript. Since Bcl2 is most important member in Bcl2 family and contains the classic BH3 domain, Bcl2 should be included in all studies as control. Additionally, study in this manuscript has limited physiological significance because most experiments are carried out in the overexpression systems. Therefore, the present version of manuscript is not suitable for publication in Nature Communications.

Specific issues:

1. It is know that majority of Mcl-1 and Bcl-XL are localized in mitochondria, only very small amount of Mcl-1 and Bcl-XL are located in ER. Authors claim that the mechanism of Bcl2 family proteins (Mcl-1 and Bcl-XL) up-regulation of α7 nAChRs via their interactions in ER. How such interaction of Bcl2 protein(s)/ α7 nAChRs in ER can significantly enhance α7 expression? A direct mechanism should be identified. Authors should test whether Bcl2 family proteins prevent E3 ligase to target α7 nAChR leading to suppression of its ubiquitination and degradation. Author should also test whether Bcl2 family proteins regulate α7 expression at transcriptional or translational level.
2. The sequence of the BH3 motif in α7 is not highly conserved with the sequence of Mcl-1, Bcl-XL, PUMA, Noxa, Bax and Bak. Among 8 amino acids in the BH3 domains, only 1 or 3 amino acids in α7 are conserved with Bcl2 family proteins, α7 vs Noxa: 1 amino acid, α7 vs PUMA: 1 amino acid; α7 vs Bak: 2 amino acids, α7 vs Bax: 3 amino acids, α7 vs Bcl-XL: 2 amino acids, α7 vs Mcl-1: 2 amino acids. It is not sure that is real BH3 motif.
3. Addition to fluorescent α-Bgt labeling, fluorescent αα7 labeling should also be included to directly detect α7 nAChR expression.
4. Most of experiments in this manuscript used overexpression systems. How about effect by depleting endogenous Bcl2 family proteins on α7 expression in various types of cells?
5. Since authors study the effect of Bcl2 family proteins on α7 nAChR expression. Bcl2 should be included in the all experiments in addition to studying Mcl-1, Bcl-XL and Bcl-W.

Reviewer #2 (Remarks to the Author):

This manuscript concerns novel findings starting from a >17,000 cDNA clone, high-content imaging screening study for entities that promote assembly and expression as ligand-binding complexes of α7-nicotinic acetylcholine receptors (α7-nAChRs). Interestingly, the screening identified anti-apoptotic, B-cell lymphoma 2 (Bcl-2) family proteins. Mutation studies show that facilitated assembly, total and cell surface expression, and functional expression of α7-nAChRs occur via a Bcl-2 homology 3 (BH3) binding domain-like motif for pro-apoptotic Bcl-2 proteins on the α7 subunit large, second intracellular loop without altering pro-expression effects of two other chaperones. Natural cell surface expression of α7-nAChRs in rat hippocampal neurons also is

elevated by lentivirus-mediated overexpression of Bcl-2 family member myeloid cell leukemia 1 (Mcl-1) but suppressed by Bcl-2 inhibitors. The results are interpreted as linking cell survival/death-regulating proteins and $\alpha 7$ -nAChR expression.

Generally, the results are interesting, and the conclusions are supported by the data. Differences between effects of Bcl-2 proteins and Ric-3 and for synergy across the three "classes" of chaperones (NACHO, Ric-3, Bcl-2 proteins) in modulating $\alpha 7$ -nAChR expression are convincing, especially when results are put into context by reference to no-chaperone or only NACHO effects alone. It certainly makes sense that nAChR family members will tie to a number of biological processes, independent of their roles as ligand-gated ion channels. Speculation now based on interactions between $\alpha 7$ -nAChR and Bcl-2 family proteins about roles of nAChR in neuronal and other cell survival is justified, although much more needs to be done to more firmly establish biological relevance of those interactions.

Minor points: In the abstract and introduction, it is suggested that the authors indicate that nAChRs both modulate and mediate synaptic transmission and are possible rather than "valuable" (unproven) therapeutic targets. Also, the broad statement that heterologous expression of functional $\alpha 7$ -nAChR is impaired in mammalian cell lines is not entirely correct, as such has been achieved, albeit perhaps reflecting basal expression of Ric-3 and/or NACHO in permissive cell hosts. The contention that Bcl-2 protein co-expression induces larger increases in total rather than cell surface levels of α -bungarotoxin-binding $\alpha 7$ -nAChR does not seem to match results shown in Fig. 1, and perhaps it requires specification of the extent of increase across conditions, as in the form of a summary table indicating differences in expression or function relative to a non-NACHO or plus-NACHO control, as pertinent. That approach is especially illuminating, showing a general 2-fold increase in $\alpha 7$ -nAChR expression induced by Ric-3 co-expression when absolute levels of expression vary considerably when nAChR $\alpha 7$ subunit second transmembrane domain modifications are done. Is an explanation warranted about why there are larger increases in epibatidine as opposed to α -bungarotoxin binding, respectively, to broken cell membrane preparations or permeabilized cells upon co-expression with Bcl-2 family proteins but not upon Ric-3 co-expression? The presentation of data and clarification of conditions used for each study could be more clear if conditions were spelled out in the results narrative and in the figure legend (e.g., Fig.2d), as it is not always easy to find what ratios of constructs were used. The manuscript would benefit from some slight improvements in clarity by distinguishing between effects on $\alpha 7$ subunits as opposed to $\alpha 7$ -nAChR. Fig. 5b left is cropped in the provided figure images.

I am, in principle, opposed to anonymity for referees of manuscripts and grant proposals, and I request that this critique be communicated to the author/applicant over my signature.

R J Lukas

Reviewer #3 (Remarks to the Author):

This manuscript by Dawe et al., describes a surprising regulation of nAChRs by several anti-apoptotic Bcl-2 family members. This discovery was the result of an unbiased genome-wide unbiased screen for novel proteins regulating nAChRs. The study is solid, convincing, and of broad interest to the field –actually cutting across disciplines to reveal an unappreciated convergence. Furthermore, the findings open up new avenues for drug discovery by targeting cell survival pathways.

I have a few suggestions to improve clarity.

1. Throughout the figures, the graphs do not have an asterisk nor comparison bars to make clear what conditions are significant, and if significant compared to what other condition.
2. Do $\alpha 7$ nAChRs co-IP with Mcl-1 and Bcl-XL. I'm not sure if there is a technical hurdle, but this

would enhance the study.

3. Do co-expression of Ric-3 and the Bcl-2 proteins synergize?

4. In figure 1a, are the images corresponding to the 5x condition indicated on graph? This should be clearer in the legend.

5. Fig 3a—the cartoon indicates that ligand is bound in the lumen of the ER. Is this the case? Also, Fig 3a, the alignment seems off—would be better for each protein to start with HVED aligned.

6. For lentivirus, the figure says 'infected', but I think 'transduced' is more accurate.

We have addressed each of the reviewers' concerns as documented below.

Reviewer #1:

The reviewer stated that our “*findings are interesting, which identified a potential new signaling pathway between Bcl2 family proteins and $\alpha 7$ nAChRs.*” However, the reviewer also felt the paper needed additional mechanistic studies, results addressing endogenous $\alpha 7$ -Bcl-2 protein interactions, and experiments incorporating Bcl-2 itself. We have addressed these general concerns and his/her specific issues as follows.

Specific issues:

1. It is known that majority of Mcl-1 and Bcl-XL are localized in mitochondria, only very small amount of Mcl-1 and Bcl-XL are located in ER. Authors claim that the mechanism of Bcl2 family proteins (Mcl-1 and Bcl-XL) up-regulation of $\alpha 7$ nAChRs via their interactions in ER. How such interaction of Bcl2 protein(s)/ $\alpha 7$ nAChRs in ER can significantly enhance $\alpha 7$ expression? A direct mechanism should be identified. Authors should test whether Bcl2 family proteins prevent E3 ligase to target $\alpha 7$ nAChR leading to suppression of its ubiquitination and degradation. Author should also test whether Bcl2 family proteins regulate $\alpha 7$ expression at transcriptional or translational level.

We appreciate the reviewer's concern that Bcl-2 protein expression in the ER is low, relative to levels found in mitochondria, raising the prospect that $\alpha 7$ nAChR assembly might be augmented by some indirect action of Bcl-2 family members. One such possibility brought forward by the reviewer is that Bcl-2 members reduce $\alpha 7$ ubiquitination and degradation, leading to an accumulation of protein, from which there are more subunits to assemble into functional channels. We tested this possibility by incubating $\alpha 7$ and Bcl-X_L or Mcl-1 co-transfected cells in high concentrations (i.e. 1 and 10 μ M) of the proteasome inhibitor MG-132 for 24 h (see Supplementary Figure 1, panel e). Similar concentrations of MG-132 have been shown to modestly increase $\alpha 4\beta 2$ nicotinic receptor assembly and expression (Govind et al., 2012), though the ubiquitination of $\alpha 7$ receptors remains poorly studied. However, the upregulation of pentamer assembly (measured by α -bungarotoxin labeling) by Bcl-2 members was preserved in the presence of MG-132, suggesting they do not act via suppression of ubiquitination or protein turnover. Moreover, if Bcl-2 proteins simply reduced ubiquitin ligase activity, single point mutations on the $\alpha 7$ intracellular loop (e.g. I436A mutation in Figure 4) should not disrupt Bcl-2-dependent upregulation, since we did not target lysine residues, which are generally required for ubiquitination.

2. The sequence of the BH3 motif in $\alpha 7$ is not highly conserved with the sequence of Mcl-1, Bcl-XL, PUMA, Noxa, Bax and Bak. Among 8 amino acids in the BH3 domains, only 1 or 3 amino acids in $\alpha 7$ are conserved with Bcl2 family proteins, $\alpha 7$ vs Noxa: 1 amino acid, $\alpha 7$ vs PUMA: 1 amino acid; $\alpha 7$ vs Bak: 2 amino acids, $\alpha 7$ vs Bax: 3 amino acids, $\alpha 7$ vs Bcl-XL: 2 amino acids, $\alpha 7$ vs Mcl-1: 2 amino acids. It is not sure that is real BH3 motif.

Molecular interactions between Bcl-2 proteins are mediated by a core set of three hydrophobic residues (often referred to as positions H2 to H4 in the Bcl-2 literature) within the BH3 domain. An examination of human (pro-apoptotic) BH3 sequences shows that position H2 universally contains leucine, whereas positions H3 and H4 can be isoleucine, leucine, methionine, phenylalanine, or valine -the exact identity is not critical. Finally, the two residues immediately preceding H4 are usually acidic (i.e. aspartate or glutamate), while other positions between H2 and H4 have great variability in terms of their chemical

composition. There is no strict requirement of sequence identity within the BH3 domain for Bcl-2 proteins to be classified as such.

In our manuscript, we have defined the $\alpha 7$, BH3-like motif based on amino acid “conservation” with equivalent positions within BH3 domains of Bcl-2 proteins. Supporting this assertion, position H2 is leucine, positions H3 and H4 are both hydrophobic (isoleucine and valine), and the residues prior to H4 are both acidic (glutamate). In this sense, five amino acids (within a span of eight) where it is possible to assess sequence conservation, are in fact conserved. Moreover, by mutating the H2, H3, and H4 equivalent positions of $\alpha 7$ (L433, I436, and V440), we were able to disrupt Bcl-2 protein-mediated upregulation of receptor assembly, as would be expected for a BH3-type interaction. Accordingly, we are confident that the preM4 helix of $\alpha 7$ is behaving like a canonical BH3 domain in the presence of anti-apoptotic Bcl-2 proteins.

3. Addition to fluorescent α -Bgt labeling, fluorescent α alpha7 labeling should also be included to directly detect $\alpha 7$ nAChR expression.

The reviewer raises a valid point in that changes in α -Bgt labeling could reflect differences in $\alpha 7$ protein expression, as opposed to increased pentamer assembly. Because we have found that many commercially available antibodies poorly label nicotinic receptor subunits, we used antibodies directed against an HA motif on the C-terminal of a tagged $\alpha 7$ construct to report subunit expression via immunostaining. New experiments included in Supplementary Figure 1 (panel f) illustrate that HA fluorescence intensity remains virtually unchanged when Bcl-2 proteins are coexpressed with $\alpha 7$ and NACHO, whereas α -Bgt fluorescence from the same cells is several-fold greater. These data further address another concern of the reviewer (see Issue #1) and show that Bcl-2 protein-mediated increases in $\alpha 7$ pentamer formation reflect enhanced $\alpha 7$ assembly rather than decreased $\alpha 7$ degradation.

4. Most of experiments in this manuscript used overexpression systems. How about effect by depleting endogenous Bcl2 family proteins on $\alpha 7$ expression in various types of cells?

We appreciate the reviewer’s suggestion that endogenous Bcl-2 family proteins should be reduced or inhibited to demonstrate a physiological regulation of $\alpha 7$ nAChR expression. Our manuscript included experiments demonstrating that Bcl-2 inhibitors reduced α -Bgt staining in cultured hippocampal neurons (Figure 5). The only other cell type where functional $\alpha 7$ channels have been carefully studied is astrocytes, and we have now added additional experiments from astrocytes (Supplementary Figure 6). These experiments show comparable reductions in the $\alpha 7$ -mediated FLIPR (i.e. functional) response after astrocytes have been incubated with Bcl-2 inhibitors.

5. Since authors study the effect of Bcl2 family proteins on $\alpha 7$ nAChR expression. Bcl2 should be included in the all experiments in addition to studying Mcl-1, Bcl-XL and Bcl-W.

Throughout our manuscript, we emphasized data from Bcl-X_L and Mcl-1 in figures, because they are both highly expressed in the central nervous system, where $\alpha 7$ nAChRs have the greatest physiological role. In contrast, Bcl-2 itself has much lower CNS expression (Genome Tissue Expression Project <https://www.gtexportal.org>), and is far less likely to interact with nicotinic receptors there. To demonstrate that Bcl-2 similarly enhances $\alpha 7$ nAChR assembly compared with other Bcl-2 members, we have added new immunostaining data to our study (see Supplementary Figure 1b). Meanwhile electrophysiological data in Supplementary Figure 2 also highlight the upregulation of functional $\alpha 7$

nAChR expression by Bcl-2 specifically. Given that Mcl-1 and Bcl-X_L are neuronal anti-apoptotic Bcl-2 family members and generally exhibited similar behavior in all of our experiments, we did not feel it was necessary to also additionally test Bcl-2 in “all” cases, especially given the limited coexpression of $\alpha 7$ and Bcl-2 in the brain.

Reviewer #2:

The reviewer wrote that our “*manuscript concerns novel findings... the results are interesting, and the conclusions are supported by the data.*” We are thankful for the overall enthusiasm of Reviewer 2 toward our manuscript, and have addressed the reviewer’s minor concerns as indicated below.

Minor points:

In the abstract and introduction, it is suggested that the authors indicate that nAChRs both modulate and mediate synaptic transmission and are possible rather than “valuable” (unproven) therapeutic targets.

We have changed the abstract text to indicate nAChRs can indeed “mediate” synaptic transmission, in addition to its modulation of other neurotransmitters. Likewise, we have changed the description of nAChRs from valuable to “attractive” targets for pharmaceutical development.

Also, the broad statement that heterologous expression of functional $\alpha 7$ -nAChR is impaired in mammalian cell lines is not entirely correct, as such has been achieved, albeit perhaps reflecting basal expression of Ric-3 and/or NACHO in permissive cell hosts.

Because the term “impaired” might imply that functional receptor expression is entirely suppressed, we now use the phrase “generally poor” to reflect that some channel activity has been measured in mammalian cells.

The contention that Bcl-2 protein co-expression induces larger increases in total rather than cell surface levels of α -bungarotoxin-binding $\alpha 7$ -nAChR does not seem to match results shown in Fig. 1, and perhaps it requires specification of the extent of increase across conditions, as in the form of a summary table indicating differences in expression or function relative to a non-NACHO or plus-NACHO control, as pertinent. That approach is especially illuminating, showing a general 2-fold increase in $\alpha 7$ -nAChR expression induced by Ric-3 co-expression when absolute levels of expression vary considerably when nAChR $\alpha 7$ subunit second transmembrane domain modifications are done.

We agree with the reviewer’s concern that it could be difficult for readers to keep track of the relative effects of Bcl-2 member coexpression with $\alpha 7$ and NACHO across many experimental conditions. Accordingly, we have added a supplemental table to our manuscript, outlining the fold change in α -Bgt labeling induced by Bcl-2 proteins and Ric-3 from experiments where such information was not provided in figures already. While the fold change in surface α -Bgt labeling does tend to be greater than that of total α -Bgt from permeabilized cells, the absolute change in total α -Bgt fluorescence intensity is still greater than that of surface α -Bgt fluorescence intensity (i.e. from Figure 1: with Mcl-1, 203 a.u. for

surface α -Bgt and 485 a.u. for total α -Bgt). We have updated the text of the results section to clarify our point, which is that Bcl-2 proteins must enhance the intracellular assembly of $\alpha 7$ pentamers, rather than simply trafficking assembled pentamers to the cell membrane (as that alone would not increase total α -Bgt staining).

Is an explanation warranted about why there are larger increases in epibatidine as opposed to α -bungarotoxin binding, respectively, to broken cell membrane preparations or permeabilized cells upon co-expression with Bcl-2 family proteins but not upon Ric-3 co-expression?

The reviewer has astutely identified that Bcl-2 member coexpression (notably Bcl-X_L) with $\alpha 7$ and NACHO produced a somewhat greater upregulation of epibatidine binding as opposed to total α -Bgt labeling. Though these techniques both reflect the extent of subunit assembly, and should therefore report broadly similar fold changes in binding sites, they do not measure identical subsets of receptors. We have found previously that ³H epibatidine binding to HEK293 membranes expressing $\alpha 7$ and NACHO cannot be entirely displaced by α -Bgt (see Matta et al. 2017), suggesting epibatidine might bind to more intermediate subunit complexes, which are not yet competent to bind α -Bgt. The formation of such complexes might be favored by Bcl-2 proteins, as opposed to Ric-3.

The presentation of data and clarification of conditions used for each study could be more clear if conditions were spelled out in the results narrative and in the figure legend (e.g., Fig.2d), as it is not always easy to find what ratios of constructs were used.

We have now made sure that the cDNA ratios transfected for each experiments are listed in the corresponding figure legends. However, to be concise we had largely omitted reporting these ratios from the results text.

The manuscript would benefit from some slight improvements in clarity by distinguishing between effects on $\alpha 7$ subunits as opposed to $\alpha 7$ -nAChR.

We thank the reviewer for highlighting this oversight, and identified a number of instances where the terms " $\alpha 7$ assembly" or " $\alpha 7$ expression" were used, which could reflect changes in individual/unassembled subunit folding or expression, respectively, as well as changes in the functional pentamer. These ambiguous terms have been clarified throughout the text, with the addition of "receptor," "pentamer," or "nAChR" after $\alpha 7$ to imply changes in receptors, or "subunit" after $\alpha 7$ to imply changes in protein expression.

Fig. 5b left is cropped in the provided figure images.

We appreciate the comment and have now fixed the figure to avoid any labels being covered.

Reviewer #3:

The reviewer characterized our work as "*solid, convincing, and of broad interest to the field.*" We appreciate the interest and commendations that Reviewer 3 had for our manuscript, and have

addressed the reviewer's suggestions as indicated below.

I have a few suggestions to improve clarity.

1. Throughout the figures, the graphs do not have an asterisk nor comparison bars to make clear what conditions are significant, and if significant compared to what other condition.

For all results, statistical tests are included in figure legends. However, in many figures we have now added indicators of a significant effect (i.e. asterisk) and/or denoted NS for non-significant effects, particularly where both outcomes occur within the same panel. In other cases, where all conditions produced robust, significant effects (i.e. Figure 1), we did not include such symbols in order to avoid extensive visual clutter within figure images.

2. Do $\alpha 7$ nAChRs co-IP with Mcl-1 and Bcl-XL. I'm not sure if there is a technical hurdle, but this would enhance the study.

Co-immunoprecipitation does face technical hurdles for this interaction. In our experience, non-specific binding typically complicates co-immunoprecipitation studies of two transmembrane proteins in transfected into HEK cells and native tissue studies are critical. Unfortunately, no available antibodies are suitable for western blotting of $\alpha 7$ in rodent brain (see Moser et al., 2007).

3. Do co-expression of Ric-3 and the Bcl-2 proteins synergize?

The reviewer brings up an interesting question in that we have contended Bcl-2 proteins facilitate $\alpha 7$ nAChR assembly through a unique mechanism from either NACHO or Ric-3. Therefore, one would expect Bcl-2 members to additionally enhance α -Bgt labeling, even in the presence of NACHO and Ric-3. We examined this possibility in new experiments and found that such synergy indeed exists. Bcl-X_L and Mcl-1 proteins each further increase $\alpha 7$ nAChR surface expression ~2.5 fold when co-transfected atop NACHO and Ric-3. These data are now presented in Supplementary Figure 1c.

4. In figure 1a, are the images corresponding to the 5x condition indicated on graph? This should be clearer in the legend.

We have updated the figure legend to indicate the cDNA ratios transfected prior to obtaining the images presented in Figure 1a. The 1:3:5 ratio of $\alpha 7$:NACHO:Bcl-2 member cDNA was used.

5. Fig 3a—the cartoon indicates that ligand is bound in the lumen of the ER. Is this the case? Also, Fig 3a, the alignment seems off—would be better for each protein to start with HVED aligned.

For clarity, we have removed the acetylcholine (ACh) molecule from panel 3a.

Meanwhile, during earlier editing of Figure 3, the alignment of sequences was inadvertently altered. This has now been corrected.

6. For lentivirus, the figure says 'infected', but I think 'transduced' is more accurate.

We have modified the labeling of the figure and related text to read “transduction” rather than “infection.”

REVIEWERS' COMMENTS:

Reviewer #1 (Remarks to the Author):

Authors have addressed my concern properly, and suggested to accept this revised manuscript in "Nature Communications".

Reviewer #2 (Remarks to the Author):

The authors have done a nice job of responding to reviewers' previous, major concerns about experimental design and interpretation. Although some improvements have been noted, there remains language imprecision and sloppiness in the presentation, and one could quibble about the sequence of data presented and depth of analysis of some of the data.

Minor points:

Strictly speaking (Introduction, first paragraph), nAChR mediate and modulate synaptic transmission, and $\alpha 9$ subunits also can form homomers, and human SH-EP1 epithelial cells are proven to be good hosts for $\alpha 7$ - nAChR expression (e.g., see reference 36). Stylistically, there are many places where the presentation still isn't precise and shortcuts are taken not specifying $\alpha 7$ "subunits" or "receptors," "receptor pentamers".

Results, first paragraph: As an example of imprecision or sloppiness of language in the narrative, if my understanding is correct, wouldn't the sentence containing ".. cotransfected with $\alpha 7$ and NACHO .." be better phrased as "We co-transfected HEK293 cells with individual plasmids encoding nAChR $\alpha 7$ subunits, NACHO and single, specific [or somehow specify the typical number of genes per pool if screening was first done with more than one test construct) constructs from a genome-wide collection of over 17,000 cDNA clones in order to identify those test constructs coding for proteins that facilitate $\alpha 7$ -nAChR expression. The latter was assessed using high-content imaging of levels of fluorescent α -bungarotoxin-labeled $\alpha 7$ -nAChR assemblies." Also, the passage " $\alpha 7$ surface labeling by α -Bgt was at least 70 % that of Ric-3" is cryptic – do the authors mean "levels of $\alpha 7$ -nAChR surface expression based on α -Bgt labeling was at least 70% of that seen on co-expression of nAChR $\alpha 7$ subunits and NACHO with Ric-3"? There are numerous passages and sentence that have these stylistic flaws requiring very careful editing and adjustments to sentence structure and content.

It is satisfying to see the contention about effects of Bcl-2 protein co-expression on total rather than cell surface levels of α -bungarotoxin-binding modified in the revised manuscript, and the supplemental summary table nicely illustrates effects normalized to relevant control samples. However, whereas the former does suggest that there are effects on subunit assembly, I still don't see why it is an either-or situation with respect to Bcl-2 proteins also facilitating trafficking to the cell surface.

It still would be warranted for the authors to explain why they think that there are larger increases in epibatidine as opposed to α -bungarotoxin binding upon co-expression with Bcl-2 family proteins but not upon Ric-3 co-expression. Is there enough α -Bgt-resistant epibatidine binding to account for the differences? There might be significance in those observations, for example, in the assembly +/- trafficking argument?

I remain a little concerned about the readability of some of the figures, especially those that are four panels wide, unless they will be shown in landscape format.

Reviewer #3 (Remarks to the Author):

The paper by Dawe et al. has been significantly improved by revisions. All of my concerns were addressed.

REVIEWERS' COMMENTS:

Reviewer #1 (Remarks to the Author):

Authors have addressed my concern properly, and suggested to accept this revised manuscript in "Nature Communications".

We are thankful for the comments of Reviewer #1, and that he or she feels we have addressed all original concerns sufficiently to warrant acceptance.

Reviewer #2 (Remarks to the Author):

The authors have done a nice job of responding to reviewers' previous, major concerns about experimental design and interpretation. Although some improvements have been noted, there remains language imprecision and sloppiness in the presentation, and one could quibble about the sequence of data presented and depth of analysis of some of the data.

We appreciate the additional comments of Reviewer #2, including that we have addressed the main concerns raised during the first round of review. Taking into account the reviewer's remaining minor points, we have made changes to the manuscript text to address concerns regarding the precision of language.

Minor points:

Strictly speaking (Introduction, first paragraph), nAChR mediate and modulate synaptic transmission, and $\alpha 9$ subunits also can form homomers, and human SH-EP1 epithelial cells are proven to be good hosts for $\alpha 7$ - nAChR expression (e.g., see reference 36).

In our abstract we indeed note that nAChRs both "mediate and modulate synaptic transmission throughout the brain." However, in the introduction we have only used the term "mediate" and not modulate, because the sentence refers specifically to cholinergic synapses.

Given that $\alpha 9$ subunits have been shown to form functional homomeric receptors when expressed recombinantly, language referring to $\alpha 7$ as "uniquely" forming homomeric complexes has been changed to note that this property of the subunit is "atypical" amongst nAChRs.

Appreciating the reviewer's comment that some mammalian cell lines facilitate reasonable expression of $\alpha 7$ nAChRs, we modified the introduction to state that "recombinant expression of functional $\alpha 7$ nAChRs is poor in **most** mammalian cell lines" and cited reference 36 here as an example where receptor activity has been detected.

Stylistically, there are many places where the presentation still isn't precise and shortcuts are taken not specifying $\alpha 7$ "subunits" or "receptors," "receptor pentamers".

We have carefully looked throughout the manuscript text for instances where $\alpha 7$ regulation or expression is written about without further specification of subunits versus (pentameric) receptors. In

certain rare instances we have been constrained by the word or character limit, such as the abstract or results section headings, and had to use very succinct language. However, in other cases where it was possible to elaborate further between subunits and receptor complexes that distinction has now been made. In a few other cases when referring to particular amino acid residues or motifs within the $\alpha 7$ protein we did not deem this distinction necessary.

Results, first paragraph: As an example of imprecision or sloppiness of language in the narrative, if my understanding is correct, wouldn't the sentence containing "... cotransfected with $\alpha 7$ and NACHO .." be better phrased as "We co-transfected HEK293 cells with individual plasmids encoding nAChR $\alpha 7$ subunits, NACHO and single, specific [or somehow specify the typical number of genes per pool if screening was first done with more than one test construct] constructs from a genome-wide collection of over 17,000 cDNA clones in order to identify those test constructs coding for proteins that facilitate $\alpha 7$ -nAChR expression. The latter was assessed using high-content imaging of levels of fluorescent α -bungarotoxin-labeled $\alpha 7$ -nAChR assemblies."

To more precisely explain the cDNA library screening strategy we employed, the first sentences of the results section have been changed to the following: "We cotransfected HEK293T cells with individual plasmids encoding the $\alpha 7$ subunit, NACHO, and one other gene product from a genome-wide collection of over 17,000 cDNA clones²⁴. High-content imaging of transfected cells was then used to identify cDNA clones encoding proteins that facilitate $\alpha 7$ nAChR expression."

Also, the passage " $\alpha 7$ surface labeling by α -Bgt was at least 70 % that of Ric-3" is cryptic – do the authors mean "levels of $\alpha 7$ -nAChR surface expression based on α -Bgt labeling was at least 70% of that seen on co-expression of nAChR $\alpha 7$ subunits and NACHO with Ric-3"? There are numerous passages and sentence that have these stylistic flaws requiring very careful editing and adjustments to sentence structure and content.

As the reviewer indicates, more explanation could be provided to explain the effect of Bcl-2 protein coexpression during our cDNA library screening, especially in relation to the positive control (i.e. Ric-3 coexpression). We have therefore modified the sentence in question to read as "for each of these clones, surface labeling of $\alpha 7$ receptors by α -Bgt was at least 70 % of the maximal levels observed when NACHO and Ric-3 were coexpressed with $\alpha 7$ subunits...".

While we have earnestly tried to address the clarity of any specific sentences that have been highlighted by the reviewer, in other cases, it is difficult to gauge what might be considered imprecise, as opposed to concise language, especially when other reviewers have not indicated such concerns.

It is satisfying to see the contention about effects of Bcl-2 protein co-expression on total rather than cell surface levels of α -bungarotoxin-binding modified in the revised manuscript, and the supplemental summary table nicely illustrates effects normalized to relevant control samples. However, whereas the former does suggest that there are effects on subunit assembly, I still don't see why it is an either-or situation with respect to Bcl-2 proteins also facilitating trafficking to the cell surface.

We agree with the reviewer that Bcl-2 proteins, in essence, facilitate surface trafficking of $\alpha 7$ nAChRs. In the first results section, we have written that "intracellular assembly of $\alpha 7$ subunits was enhanced, rather than merely membrane trafficking" because our data do not point toward Bcl-2 proteins acting uniquely to augment receptor surface expression. Since total α -Bgt and epibatidine binding both increase in the presence of Bcl-2 members (in addition to surface α -Bgt labeling), it is probable that their

central influence is over pentamer assembly, which in turn allows forward trafficking to the membrane.

It still would be warranted for the authors to explain why they think that there are larger increases in epibatidine as opposed to α -bungarotoxin binding upon co-expression with Bcl-2 family proteins but not upon Ric-3 co-expression. Is there enough α -Bgt-resistant epibatidine binding to account for the differences? There might be significance in those observations, for example, in the assembly +/- trafficking argument?

From Figure 1c, the average increase in total epibatidine binding when Mcl-1 or Bcl-X_L is coexpressed atop α 7 and NACHO is between 3-4 fold. In Supplementary Table 1, the fold change in total α -Bgt labeling reported for coexpression of Bcl-2 proteins atop α 7 and NACHO (from Figure 1b) is either 3.5 ± 0.3 (Mcl-1) or 2.4 ± 0.2 (Bcl-X_L). Thus, this discrepancy is only apparent for Bcl-X_L but still not particularly striking, given that the data are from one representative experiment. To this point, Bcl-X_L induced a 2.8 ± 0.04 fold change in total α -Bgt labeling for experiments reported in Supplementary Figure 1f. One could hypothetically make the case that Bcl-X_L promotes the assembly of partially assembled subunit complexes that bind epibatidine but not α -Bgt (our previous rebuttal letter describes evidence that epibatidine and α -Bgt bind to somewhat different subsets of receptors). However, given the inability of our experiments to address/measure partially assembled complexes, we do not feel we have sufficient evidence to advance this hypothesis.

I remain a little concerned about the readability of some of the figures, especially those that are four panels wide, unless they will be shown in landscape format.

We have adhered to the numerous helpful suggestions provided by the editor to ensure that our figures conform to journal guidelines for clarity and readability.

Reviewer #3 (Remarks to the Author):

The paper by Dawe et al. has been significantly improved by revisions. All of my concerns were addressed.

We are grateful for the comments of Reviewer #3, and that he or she feels we have generated an improved manuscript that addressed all concerns.